# Digital-Twin-Based Flexibility Wires of a Circular Connector Automatic Disassembly Process

**Yu-Ren Lin, Yu-Cheng Lin and Chao-Ching Ho ***

Department of Mechanical Engineering, National Taipei University of Technology, 1, Sec. 3, Zhongxiao E. Rd., Taipei 10608, Taiwan
* Correspondence: hochao@ntut.edu.tw; Tel.: +886-2-2771-2171 (ext. 2020); Fax: +886-2-2776-4889

**Abstract:** Circular connector four-core wires are created by winding in the cable; however, wires of different colors cannot be distributed fixedly within the cables. The exact positions of the four wires inside the cable cannot be determined because the cables are stripped. In addition, the core wire is flexible and can easily bend in any direction because of the external force interference. Therefore, fully automating this process is difficult. This study aimed to employ finite element analysis software to simulate the flexibility of the core wire in a four-core cable and to design a separate module that only uses a single-axis robot and a stepper motor. The core wires wound within the cable were separated by the designed separate module, which facilitated the subsequent welding process, improved the production efficiency, and reduced the labor costs.

**Keywords:** precision machining; finite element package; shape optimization

## 1. Introduction

Connectors are important components of electronic products used in industrial development. They are responsible for transmitting analog or digital signals, and act as a bridge for information transmission between electronic products. In recent years, connectors have been widely applied in the fields of artificial intelligence, robotics, electric vehicles, 5G communication, Industry 4.0, and smart homes. Therefore, the demand for circular connectors has significantly increased [1]. Although a circular connector automatic welding machine has been developed in the market, there are still two processes that should be completed before the welding operation. The first step is to strip the PVC insulation of the cable, and the second step is to align the cores according to the manufacturing specifications, so that they are ready to be soldered to the connector. However, the second step still requires considerable labor; in particular, the operator should separate the circular-connector core wires before welding and place them into the welding jig in the specified order. However, the disadvantages of manual operation include low consistency, efficiency, and productivity. Therefore, robotic automation operations are gradually being used to replace the operators. For example, in [2], robots were used to realize automatic assembly lines, and in [3], connector pairing in an intelligent model diagnosis robot wire harness assembly system was realized. In [4], tools and strategies for robotized switchgear cabling were demonstrated. In all of these cases, the robot is controlled by a rigid body; however, the core wire of the circular connector is flexible. Hence, realizing automatic operation for the fabrication of circular connectors is challenging. X. Li and C. C. Cheah proposed a regional feedback control method in [5], as well as the task-space sensory feedback control of robot manipulators in [6]. J.-J. Slotine and Weiping proposed an adaptive controller with a better tracking accuracy than either a PD controller or computed-torque schemes [7]. G. Niemeyer and J.-J. Slotine proposed an adaptive tracking control scheme for robots with unknown kinematic and dynamic properties [8]. Takegaki and Arimoto proposed in [9] a new feedback method for the dynamic control of manipulators. When the robot

operates winding core wires, the position and shape of the core wires are easily changed by external forces, making handling these wires with conventional robot-manipulation technology difficult [10]. X. Li, X. Su, and Y.-H. Liu [11] proposed a vision-based robotic manipulation technique for soldering flexible PCBs, which enables robots to automatically contact and actively deform flexible PCBs into desired configurations. In addition, X. Li, X. Su, Y. Gao, and Y. H. Liu proposed a new vision-based controller for robotic grasping and manipulation of USB wires in [12,13]; a two-layer structure embedded in the controller was developed. However, the experimental assumption was that the USB cores were separated and sorted.

Disassembly sequence planning is considered an NP-hard combinatorial optimization problem [14]. A digital twin refers to the virtual and computerized counterpart of a physical system and can be used to simulate the disassembly process. [15]. In [16], a framework of digital-twin-based industrial cloud robotics is implemented for robotic disassembly tasks. The modeling and simulation capabilities of digital twins enable production lines to rapidly conduct variant design and solution validation, reducing variant design time and research costs [17]. The digital-twin model provided a new technical approach for the flexible efficient and extensible assembly process of a complex product to achieve smart assembly. The purpose of this research is to employ the finite element analysis software Abaqus to simulate the flexibility of the core wires in the circular connector cable, employ the computer-aided design software Solidworks to design the separate jig, and design a separate module with a simple mechanism. Subsequently, a 3D printer was used to convert the design into a solid structure to verify the separator module. Finally, Python and Arduino software were implemented to automate the preprocessing of the soldering operation.

## 2. Background

The U.S. Department of Defense created the original specifications for military-grade circular connectors in the 1930s in order to meet the demand for sophisticated tactical and aeronautical applications. These connectors are known as type AN (for Army-Navy) connectors. The original specifications set standards for various military circular connectors. In this study, a four-core cable was selected as the research object. The core wires were sorted before welding the cable and connector. The sorting process consists of two steps. The first part is the separation operation whereas the second part is the core wire sorting operation, as shown in Figure 1. The separation operation separates the core wires from each other, and then classifies and fixes the wires according to the specifications in the separation operation. In 2018, the U.S. Patent [18] realized a method of separating the wire from the winding state; however, this method is expensive and requires a complex design of the mechanism. Therefore, in this research, we employed Abaqus to simulate the flexibility of the core wires, and Solidworks to design the separate jig. After many simulation experiments, the most effective separation fixture was designed. Thereafter, the 3D printing method was used to manufacture the separate jig, which is difficult to process with traditional machining techniques. Separate features were used to perform separation operations to achieve an economical and efficient separate module. The sorting wire process can be realized through the research of [19]; then, the preprocessing part before core wire welding can be achieved.

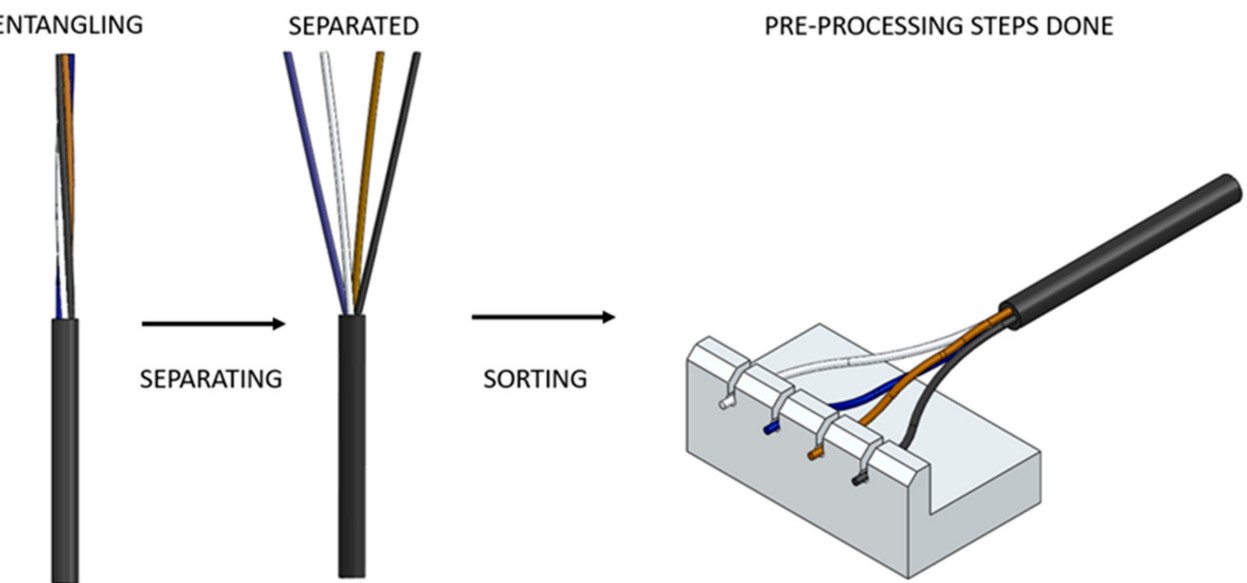

**Figure 1.** Preprocessing steps before core wire soldering include wire separation and wire sorting.

## 3. Separate Module

This section describes the overall hardware configuration of the automatic separate module designed for the separate jig. From the circular-connector core wire simulation to the separate jig and the overall machine design, including the use of a single-axis-robot, stepper motor, camera, fixed jig, motor driver, power supply, and Arduino control board.

### 3.1. Design of the Separate Jig

This section describes the design of the separation jig model using the computer-aided design software, SolidWorks. Because the separate jig manufacturing method is difficult to manufacture using general traditional processing methods, it is integrally formed using 3D printing technology, as shown in Figure 2.

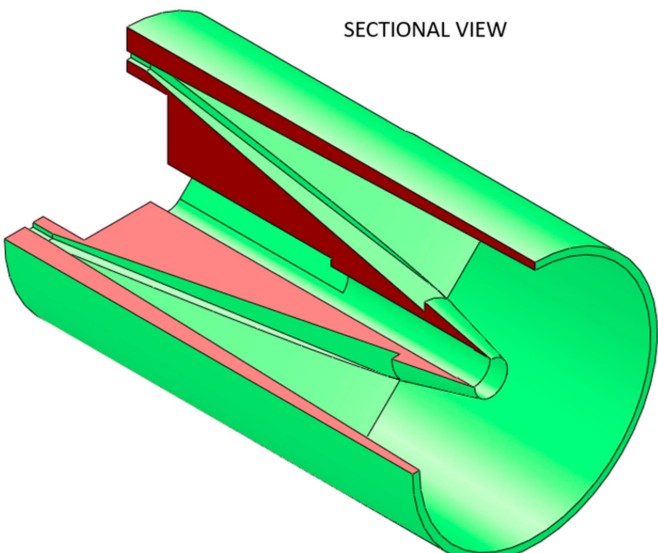

**Figure 2.** Designing a separate jig that is difficult to manufacture with traditional machining techniques.

The feature design of the separate jig is divided into three parts: the first part is the cone that extends the core wires outward; the second part is the inner cone that limits the

extension range of the core wires; the third part is the separate feature that separates the core wires from each other, as shown in Figure 3.

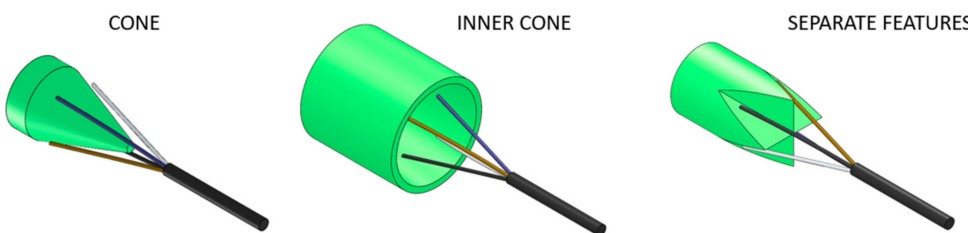

**Figure 3.** Three features of separate jig to separate wires.

Owing to the flexibility of the wire, it rebounds after being separated by the separator jig and is difficult to shape. With moving the separate jig, the axial forces exerted to the core wires due to bending moments exceed the friction and radial force exerted by the insulation of the cable and the core wires start to be separated from each other. The axial forces $F_a$ is the function is given by

$$F_a = \sum_{i=1}^{4} F_{m,i}\left(\mu_p, F_{rad}\right) \tag{1}$$

and the function variables $\mu_p$ is the inter-conduction friction coefficient and $F_{rad}$ is the radial force exerted by the insulation of the cable.

### 3.2. Simulation of Wire with Separate Jig

The finite element analysis software Abaqus was used to simulate the flexibility of the core wire in the circular connector cable. The circular-connector core wire contains two materials: PVC sheath and copper wire; however, the copper wire affects the flexibility of most core wires. Therefore, to simplify the experimental simulation results, the core wire material was set to have copper wire characteristics, that is, the core wire density was set to 8.96 g·cm−3, Young's modulus of the elastic properties was 120 GPa, and Poisson's ratio was 0.34. Thereafter, the environment and space model setup were performed. It included the separate jig, four core wires and their relative positions, degrees of freedom of the core wire in the virtual space, and relative motion between the core wire and separate jig, as shown in Figure 4.

| Core wire characteristics | |
|---|---|
| Characteristics | Value |
| Density | 8.96 g·cm$^{-3}$ |
| Young's modulus | 120 Gpa |
| Poisson's ratio | 0.34 |

| Boundary condition | | | | | | | |
|---|---|---|---|---|---|---|---|
| Region | Type | X | Y | Z | Pitch | Yaw | Roll |
| End of core wires | encastre | 0 | 0 | 0 | 0 | 0 | 0 |
| Separate-jig | displacement | 0 | 0 | 70 | 0 | 0 | 0 |

**Figure 4.** Parameters for the environment and space model setup using Abaqus.

Through the built-in finite element analysis method of Abaqus, the mesh of the core wire and the separate jig was automatically generated through a simple setting; the final setting was completed. The effect of the core wire separation was simulated, as shown in Figure 5.

The simulation results are presented in Table 1. According to the simulation results, it can be determined that if the cone angle is considerably small, the core wire will not be separated, and if the angle is significantly large, the core wire will be bent, as shown in Figure 6. For the factors affecting the success of the experiment, such as friction coefficient, relative speed, and friction force, after repeated verification, the cone is designed and optimized to improve the success rate of the separation module, as shown in Figure 7.

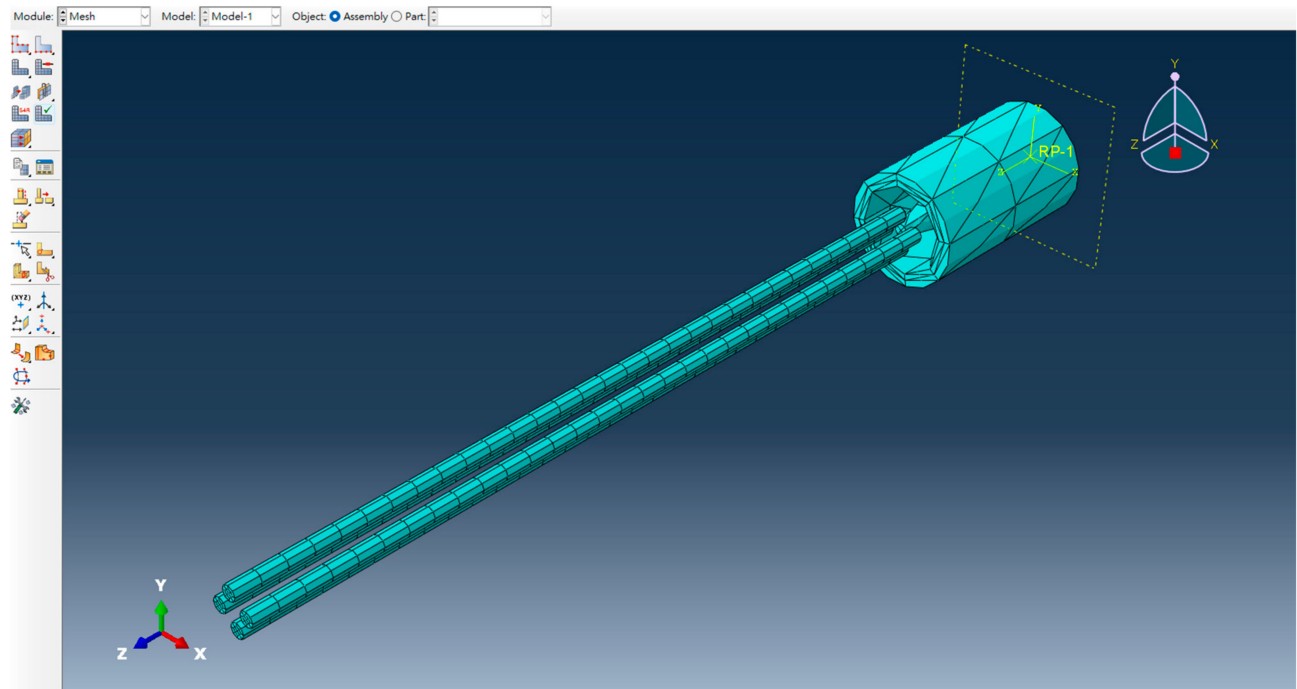

**Figure 5.** Generated mesh to perform the finite element analysis using Abaqus.

**Table 1.** Arrangement of factors affecting the separation experiment results.

| Cone Type | Cone Angle (°) | Circle Diameter (∅mm) | Friction Coefficient | Speed (mm/s) | Final Deformation (mm) |
|---|---|---|---|---|---|
| Cone A | 12 | 20 | 0.33 | 2 | 1.38 |
| Cone B | 14 | 20 | 0.33 | 2 | 1.55 |
| Cone C | 14 | 40 | 0.33 | 2 | 1.96 |
| Cone D | 15 | 40 | 0.33 | 5 | 2.25 |
| Cone E | 15 | 50 | 0.33 | 5 | 3.15 |
| Cone F | 16 | 50 | 0.33 | 5 | 3.75 |

Experiments have shown that increasing the hole spacing can better separate the wire from its shape. In addition, the advancing speed, hole size, and cone angle of the separation jig can easily squeeze the wire because of its flexibility and friction, causing the experiment to fail. The experimental parameters and design processes are presented in Table 2. The separate jig design and simulation are shown in Figure 8. Figure 9 illustrates the separated jig simulation results.

**Table 2.** Simulation results of design and experimental parameters.

| Simulation Results of Design and Experimental Parameters | | | | | |
|---|---|---|---|---|---|
| Type | Speed (mm/s) | Hole Size (∅mm) | Cone Angle (°) | Hole Spacing (mm) | Simulation Results |
| A | 2 | 16 | 12 | 5 | NG |
| B | 2 | 2 | 14 | 8 | NG |
| C | 2 | 2 | 14 | 21 | NG |
| D | 5 | 2 | 15 | 24 | NG |
| E | 5 | 2 | 16 | 24 | NG |
| F | 5 | 2 | 16 | 48 | OK |

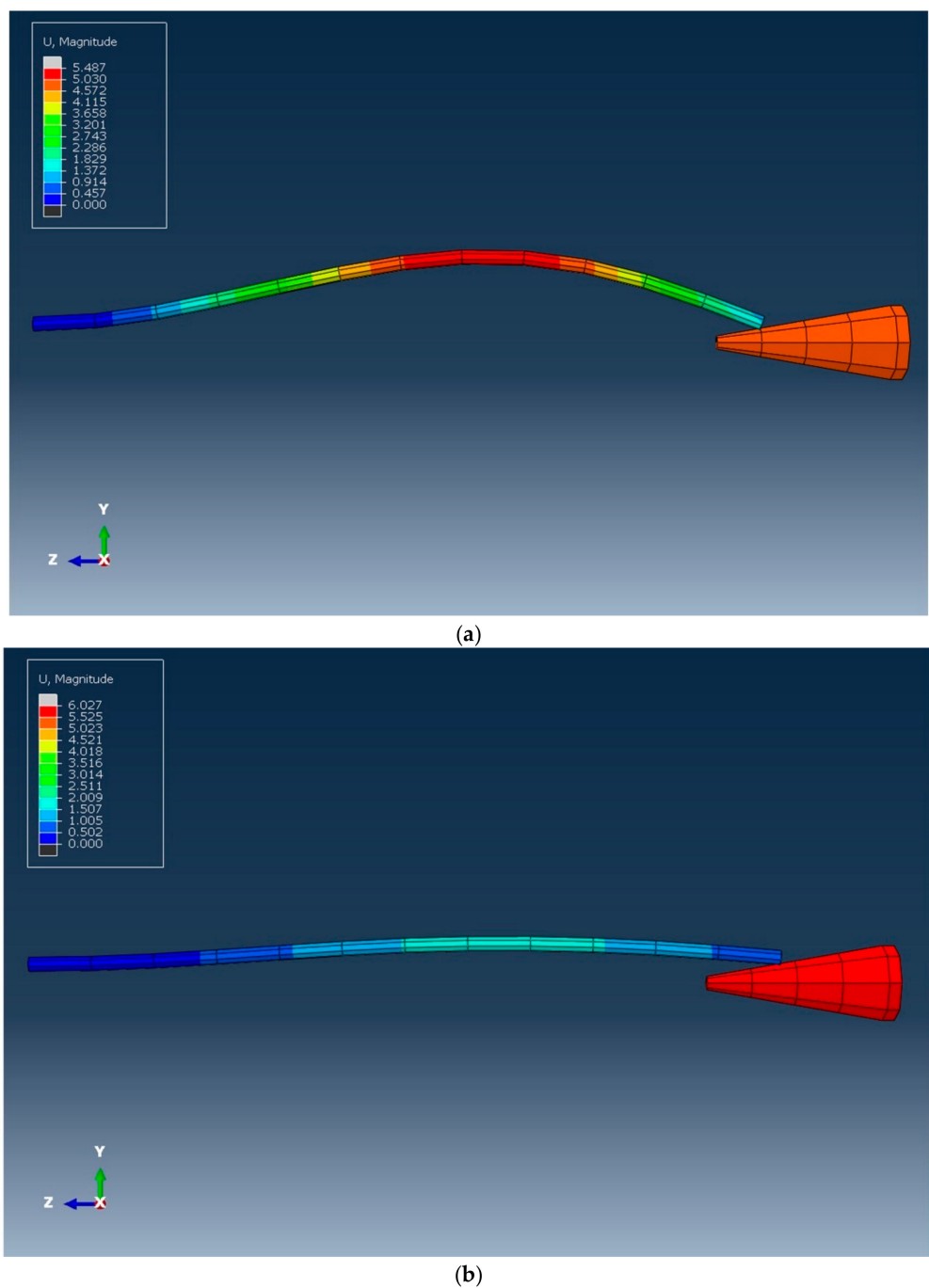

**Figure 6.** (**a**) The core wire will be bent, assuming the angle is significantly large, the relative velocity is considerably fast or the friction band is significantly large, etc. (**b**) The core wire will not be separated, assuming the angle is considerably small, the circle diameter is considerably small, etc. (units: mm).

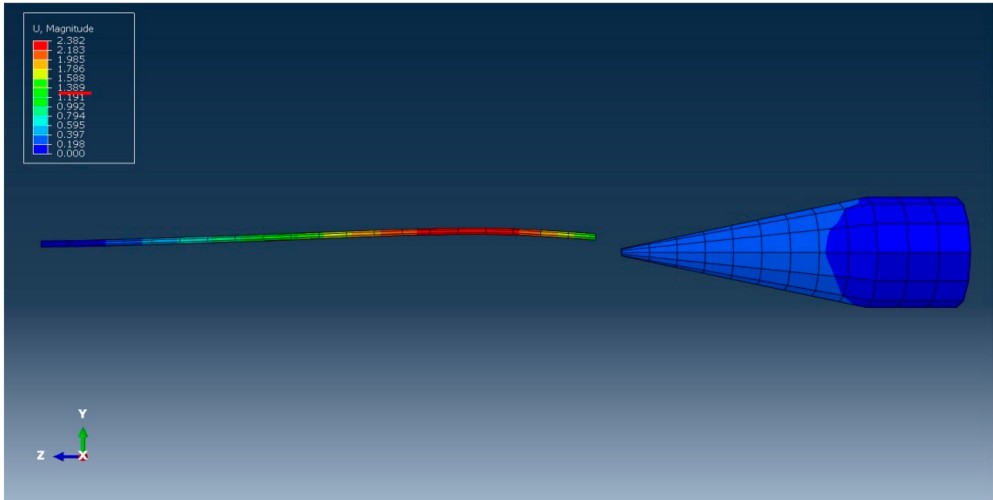

(**a**)

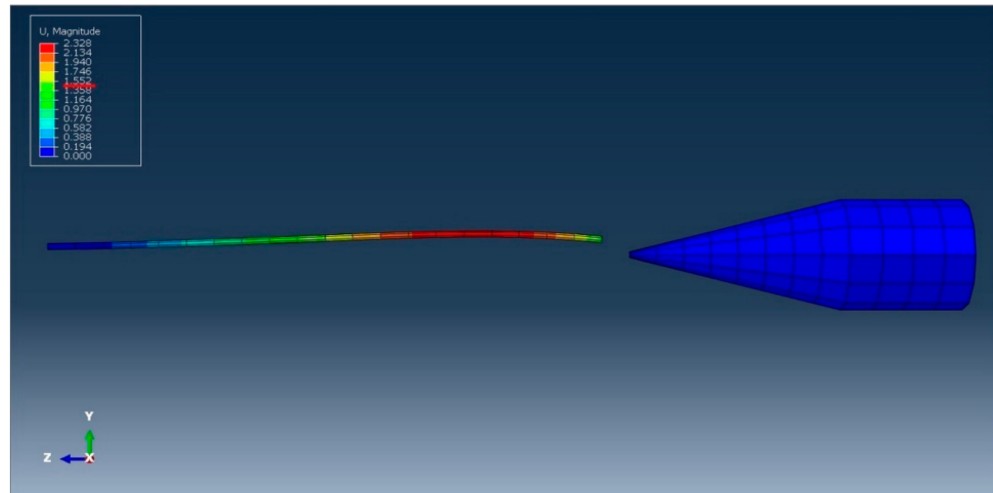

(**b**)

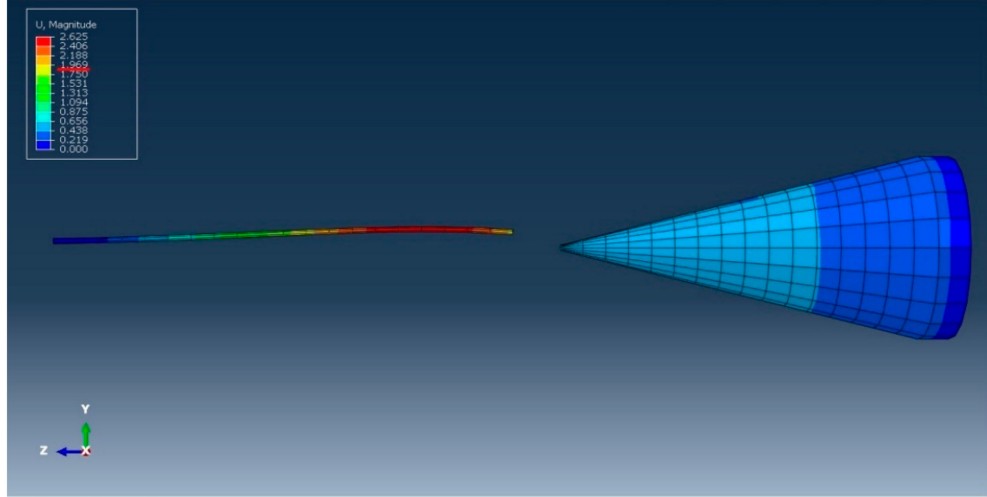

(**c**)

**Figure 7.** *Cont.*

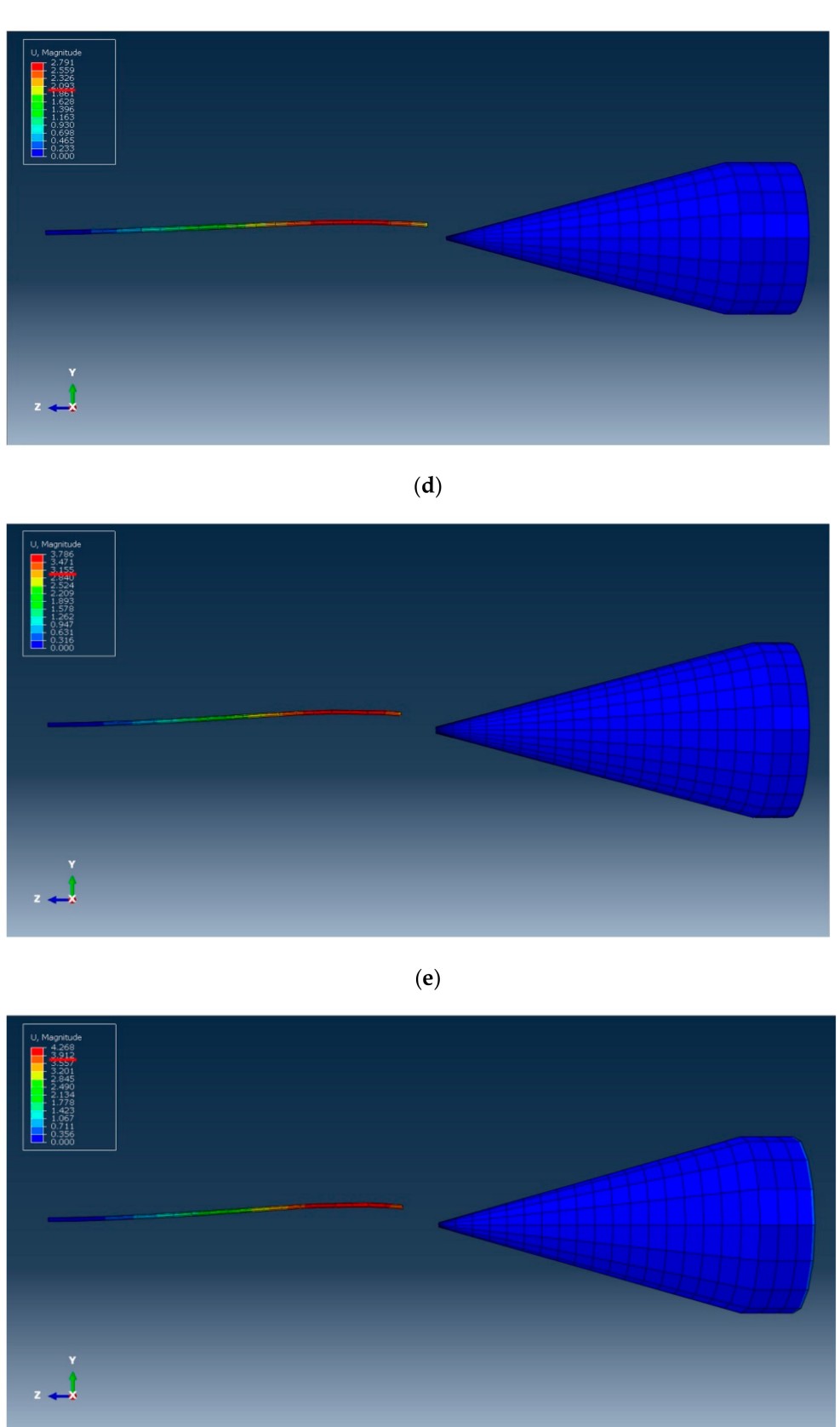

**Figure 7.** Cone design simulation results: (**a**) deformation of cone A: 1.38 mm; (**b**) deformation of cone B: 1.55 mm; (**c**) deformation of cone C: 1.96 mm; (**d**) deformation of cone D: 2.05 mm; (**e**) deformation of cone E: 3.15mm; (**f**) deformation of cone F: 3.75 mm.

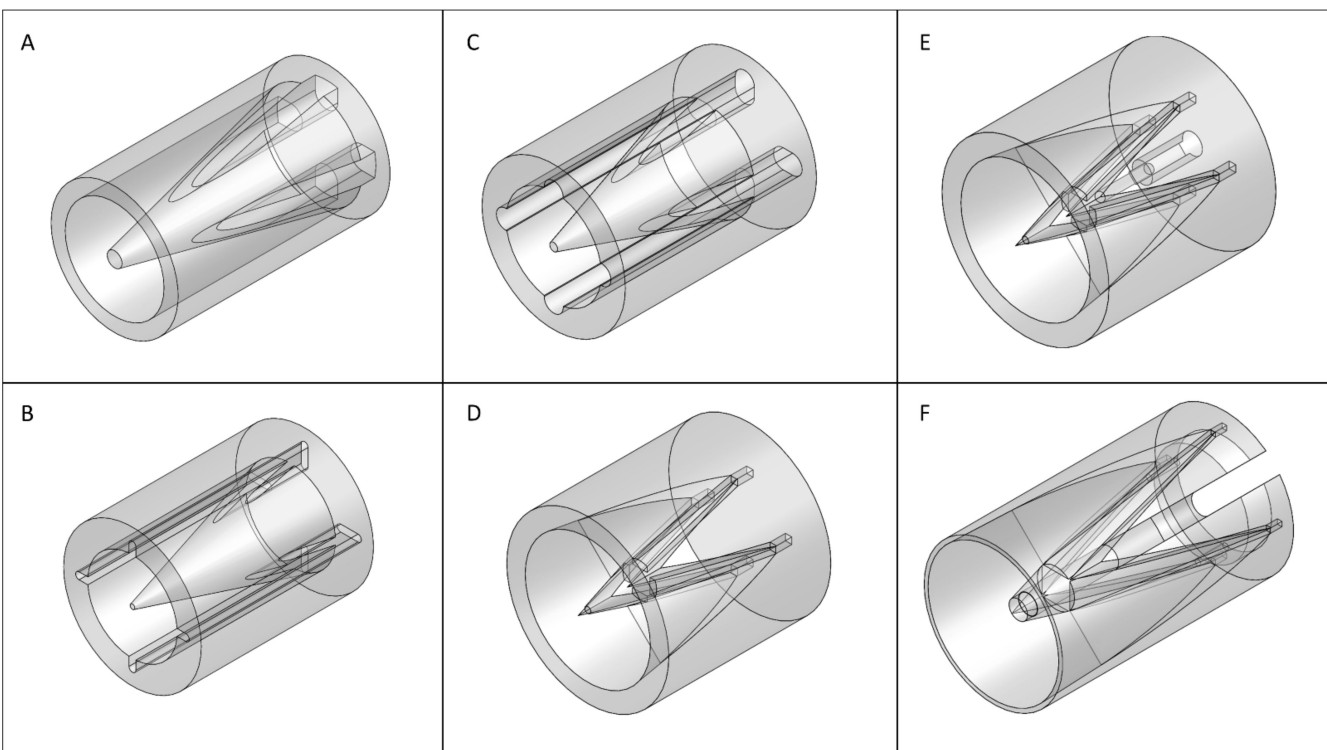

**Figure 8.** Improvement in the Separate-jig design process and analysis simulation: (**A**) type A; (**B**) type B; (**C**) type C; (**D**) type D; (**E**) type E; (**F**) type F.

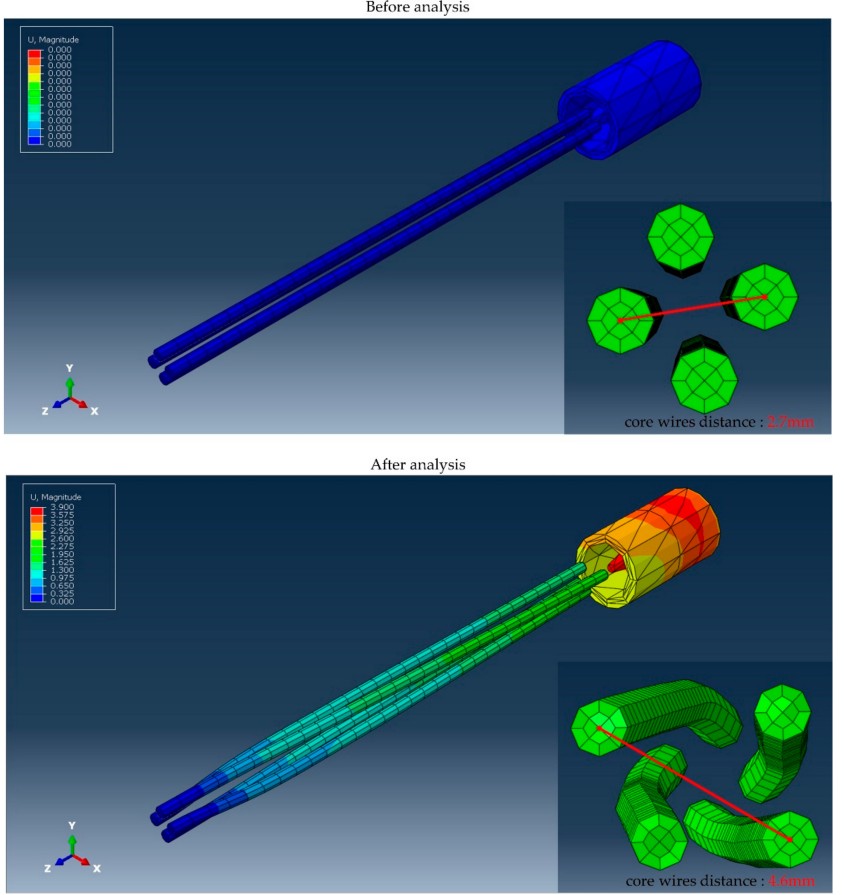

**Figure 9.** Separated jig simulation results.

*3.3. Mechanism Design*

　　This section describes the design of the separate-module equipment using computer-aided design software SolidWorks, as shown in Figure 10. The equipment list is as follows.

- Single-axis-robot: NEMA 23HS7430
- Stepper motor: NEMA 17 with encoder
- Camera: USB Endoscope Inspection Camera
- Power supply: MEANWELL-LRS-350-12
- Arduino mega board: MEGA2560 R3
- Stepper motor driver: TB6600 stepper motor driver.

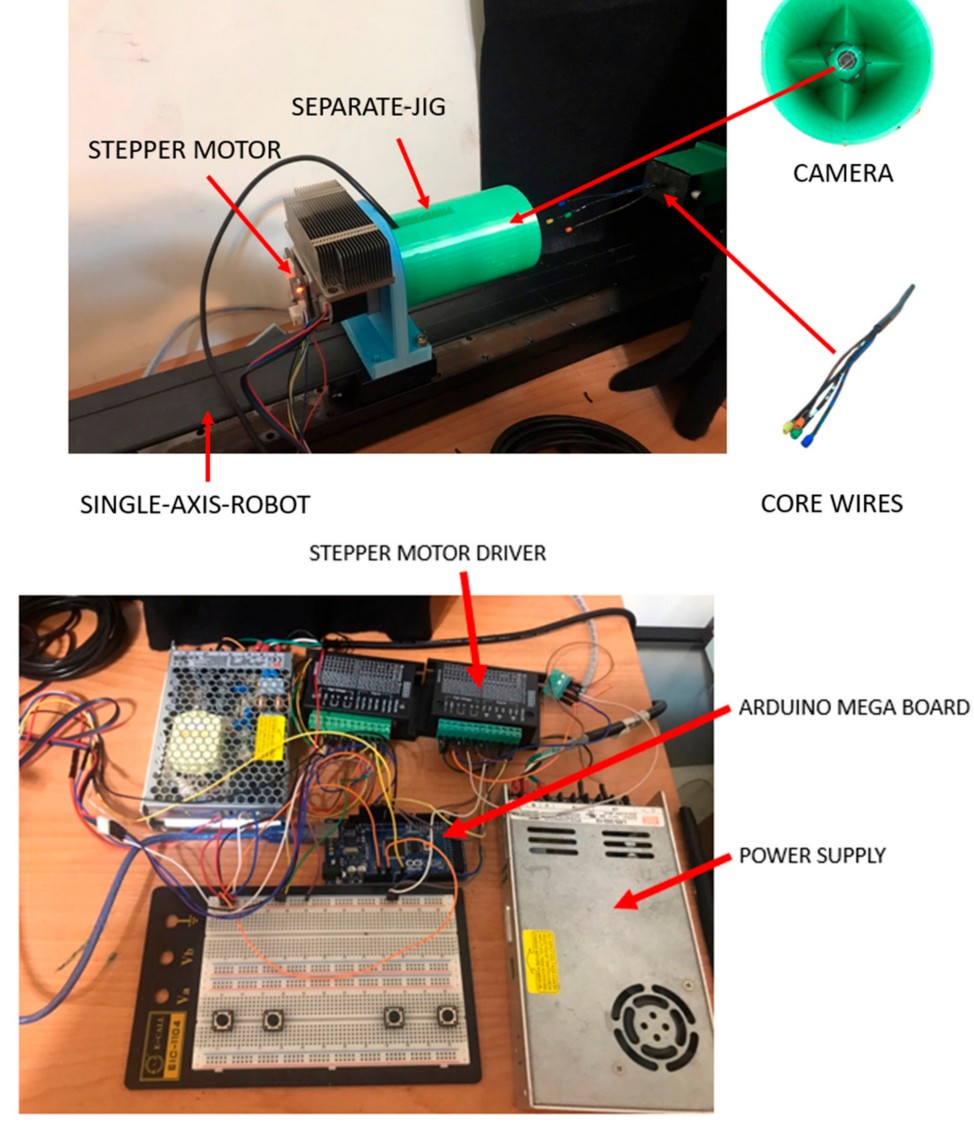

**Figure 10.** Mechanism design and all of the separate-module hardware devices.

　　When assembling, a cross line is drawn on the screen, and it is used to align the separate feature in the separate jig to control the motor to perform the separation line operation, as shown in Figure 11.

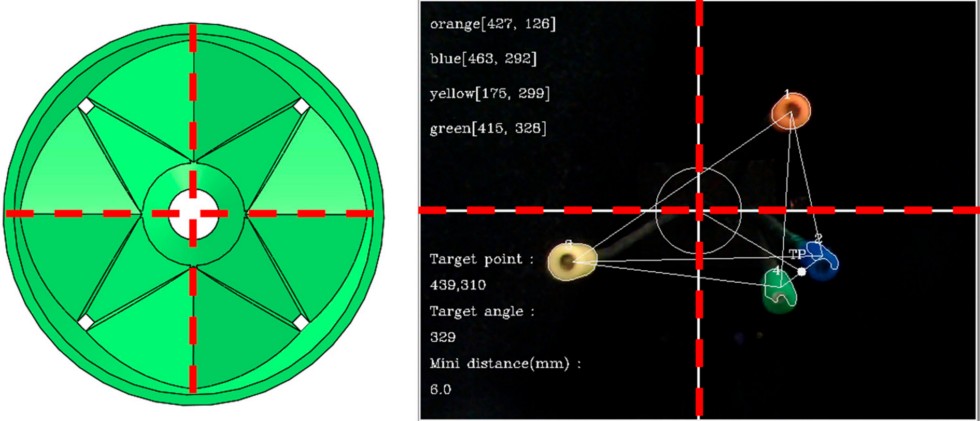

**Figure 11.** Aligning the separate features with the cross line on the screen.

*3.4. Control Scheme of the Wires Separating*

This section describes the steps involved in the wire separation process. After the hardware equipment was erected, the ratio of the pixel to the actual size could be deduced as 1 pixel = 0.67 mm. Rotational motor specification 1 pulse = 0.14°.

- In the first step, four coordinates, A, B, C, and D, were obtained by image processing, as shown in Figure 12.

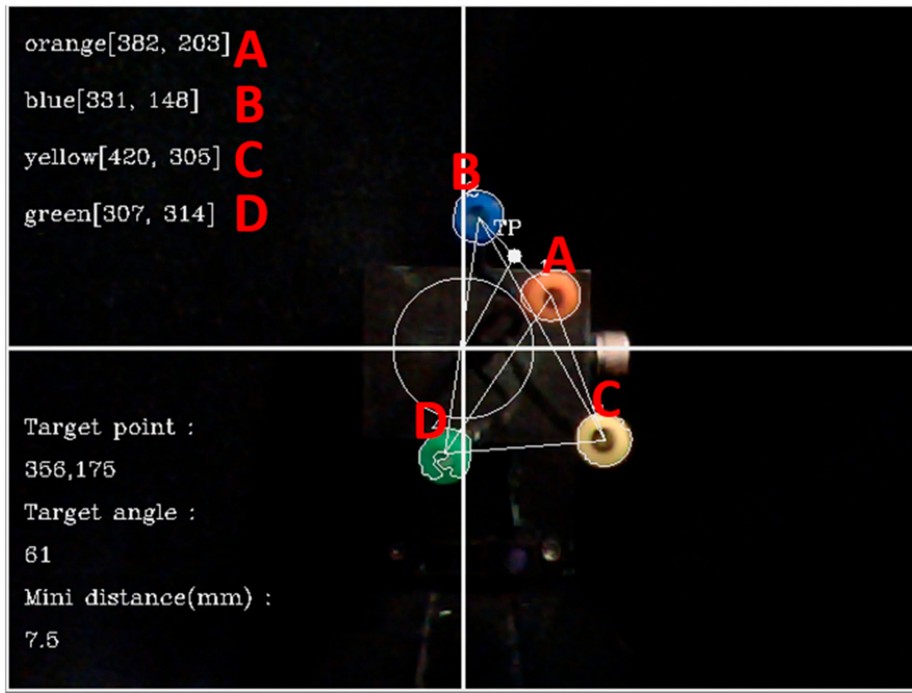

**Figure 12.** Finding 4 coordinates (i.e., A, B, C, D) in the screen by image processing.

$$D_{ab} \sqrt{(x_a - x_b)^2 + (y_a - y_b)^2} \qquad (2)$$

- In the second step, six lines could be drawn between the four coordinates, $D_{ab}$, $D_{ac}$, $D_{ad}$, $D_{bc}$, $D_{bd}$, and $D_{cd}$, as shown in Figure 13, and the distance between the lines could be found by Equation (2).

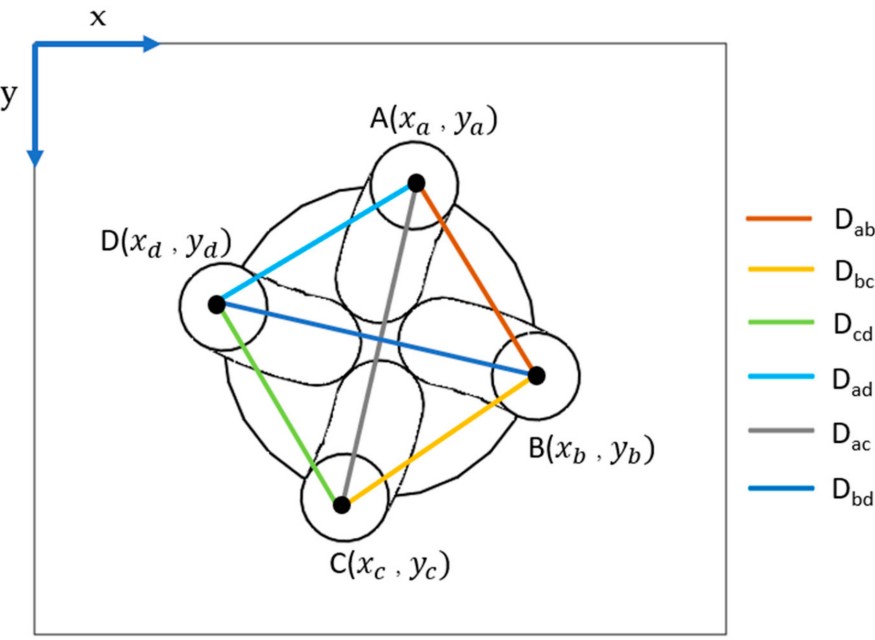

**Figure 13.** The six lines (i.e., $D_{ab}$, $D_{ac}$, $D_{ad}$, $D_{bc}$, $D_{bd}$, and $D_{cd}$) are drawn according to the four coordinates (i.e., A, B, C, and D).

- In the third step, according to the calculated length, the shortest line was found to be the smallest distance between the four core wires.

$$(x_t, y_t) = ((x_a + x_b)/2), (y_a + y_b)/2) \tag{3}$$

In the fourth step, the center point (i.e., TP) of the shortest line segment was found, as shown in Figure 14. The coordinates of the target point (TP) could be determined using Equation (3).

$$\theta = \tan^{-1}\frac{\Delta y}{\Delta x} \tag{4}$$

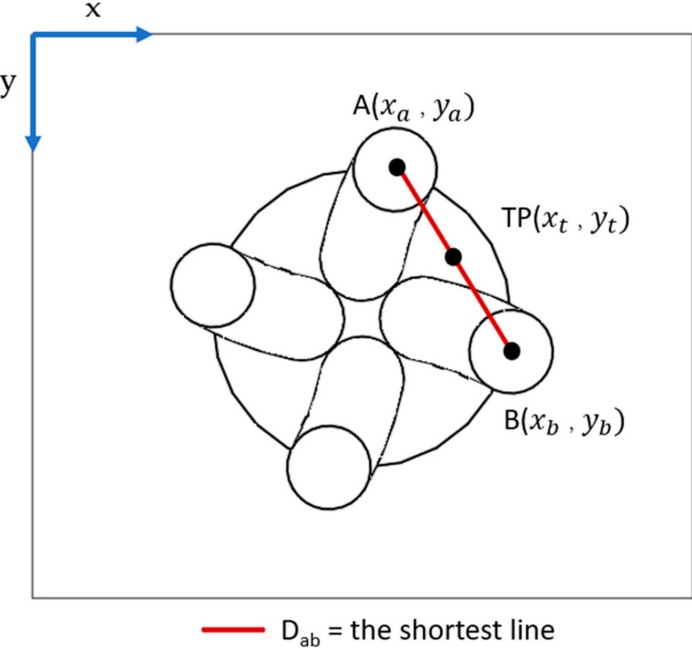

**Figure 14.** Find the center point TP of the shortest line segment.

- The fifth step was to calculate the angle between the target point and the separate feature using Equation (4), as shown in Figure 15.

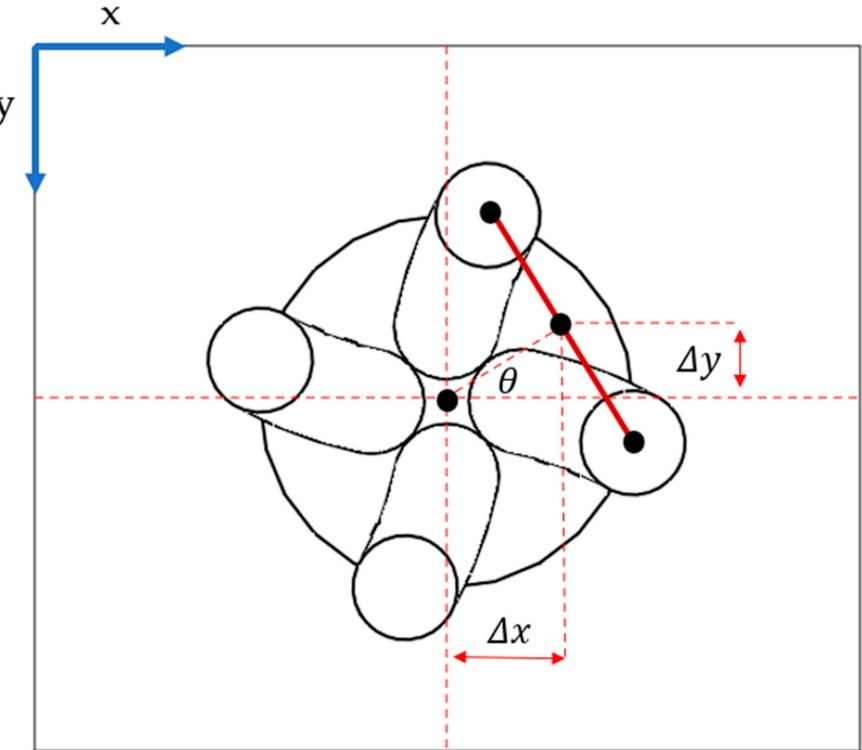

**Figure 15.** Find the angle $\theta$, which is the angle taken from the horizontal axis to the line connecting the TP and the center point.

- Sixth step: assuming that $0 < \theta < 90°$, that is TP is in the first quadrant, the rotary motor assumes the 90° line as the target to take clockwise rotation until the split line feature overlaps with TP, as shown in Figure 16. The calculation method of TP in the four quadrants is as shown in Table 3.

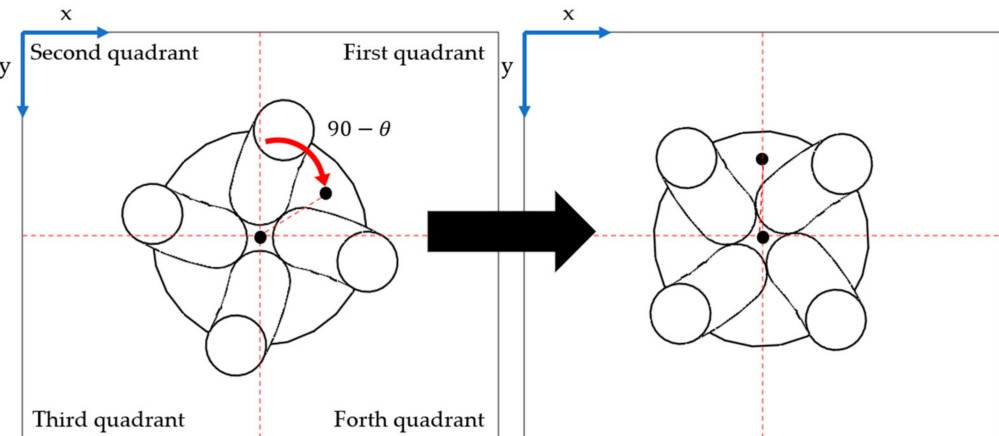

**Figure 16.** Find the angle from the vertical axis to the line connecting the TP and the center point.

**Table 3.** Calculate the pulse required by the motor according to the position of the TP point.

| Quadrant | 1st Quadrant | 2nd Quadrant | 3rd Quadrant | 4th Quadrant |
|---|---|---|---|---|
| $\theta$ | $0° < \theta < 90°$ | $90° < \theta < 180°$ | $180° < \theta < 270°$ | $270° < \theta < 360°$ |
| pulse | $(90 - \theta)/0.14$ | $(180 - \theta)/0.14$ | $(270 - \theta)/0.14$ | $(360 - \theta)/0.14$ |

- Final step: Once the separate feature is aligned between the two shortest lines, the separate jig can separate the core wire forward, as shown in Figure 17.

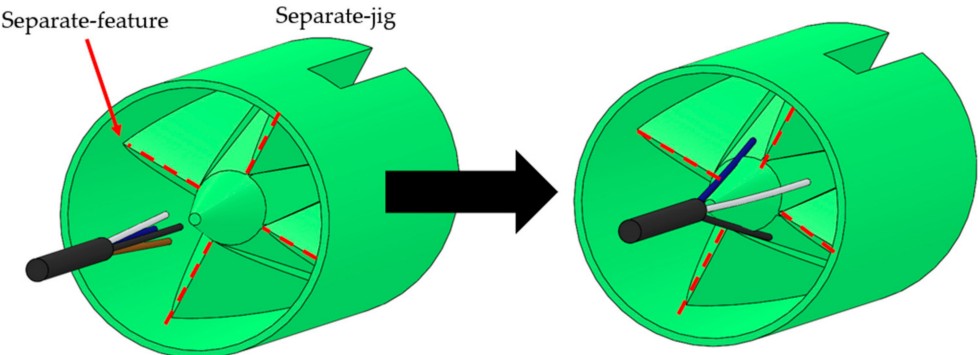

**Figure 17.** Separate the lines with a separation jig aligned with the TP point.

## 4. Experiment

This study uses a single-axis robot and an Arduino control board to design a simple and convenient tool for separating circular-connector core wires. First, Abaqus was used to simulate the flexibility of the core wire, and the simulated characteristics were used to design and optimize the separation jig. The most suitable separation jig for the core wire separation was designed by writing a set of image-processing programs in Python. Finally, the above steps were integrated to complete the automatic separation of the core wire.

### 4.1. Experimental Design

In this study, to control the motor to perform automated operations, we used the programming platform Visual Studio Code to write Python programs, and design the human–machine interface before the experiment, as shown in Figure 18. The part in the red frame in the figure can control the single-axis robot and stepper motor. The STOP button can forcefully end all programs, regardless of whether the stepper motor or single-axis-robot is in motion, that is, it will stop and end the program to ensure the safety of the machine. The part in the yellow frame in the figure is the real-time image display area, and the core wire coordinates and information required for separating the core wires are displayed in this area. The green frame in the figure is the display area of the separated core wire status, and the separated core wire button is used to execute the automatic core wire separation operation. The experimental process is illustrated in Figure 19.

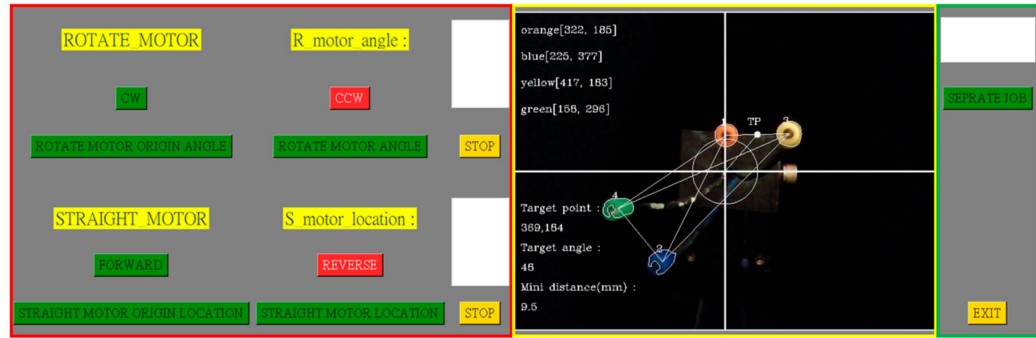

**Figure 18.** Designing a separate module GUI using Python.

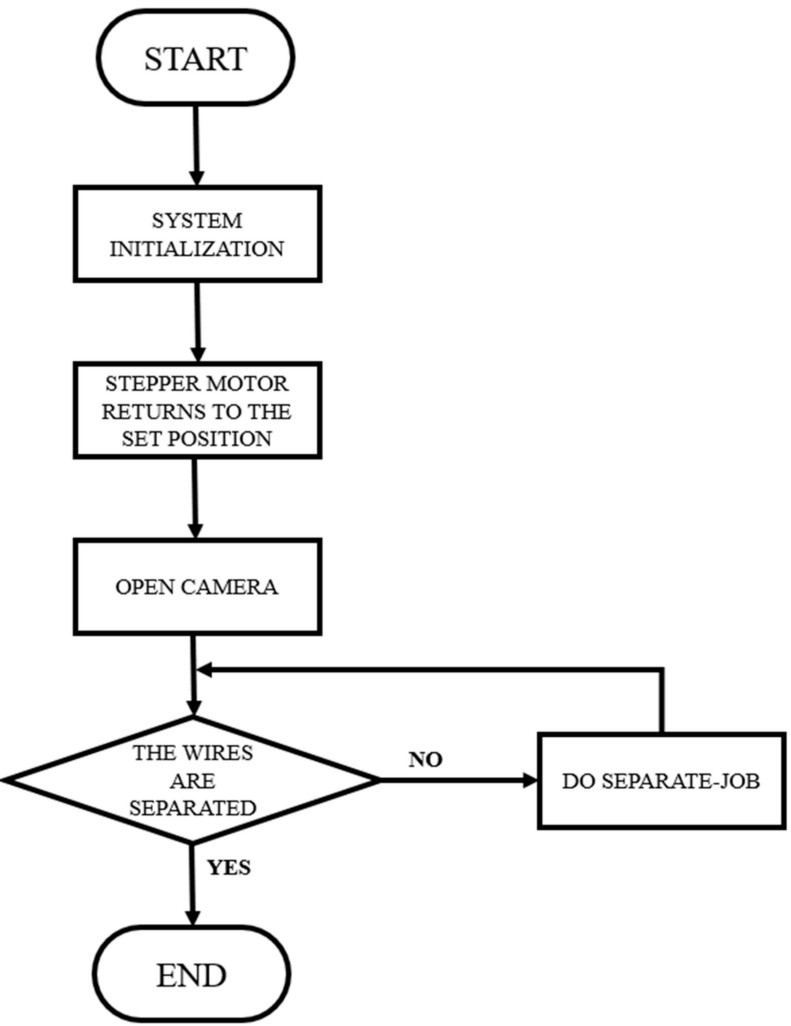

**Figure 19.** Flowchart of the automatic separation procedure.

### 4.2. Wire Separating Process

It is necessary to identify the ends of the core wires within the shooting range of the lens, and to then allocate coordinates for them. The coordinate values are used to calculate the shortest distance between each of the four core lines. The motor is controlled to align the separate feature of the separate jig with the center point of the shortest distance between the endpoints of the core wires to perform the separate job. The separate job process is illustrated in Figure 20.

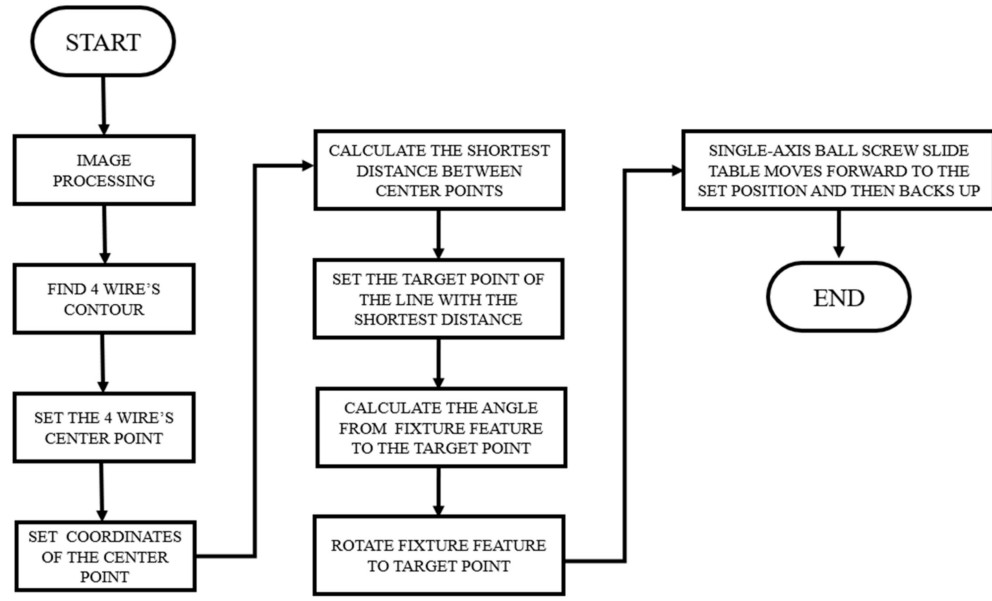

**Figure 20.** Separate job flow chart.

## 5. Experimental Results

This section first defines the experimental goal and the success or failure of the wire-separating process.

- Experimental goal: The distance between the four core wires is greater than 9 mm, which allows the robot fingers to operate on the welding platform.
- Experimental success: Rotate the fixture feature to the target point. The distance between the four core wires is greater than 9 mm.
- Experimental failure: During automatic operation, the core wire is bent out of the camera range by the separation jig.

The actual experimental screen is shown in Figure 21.

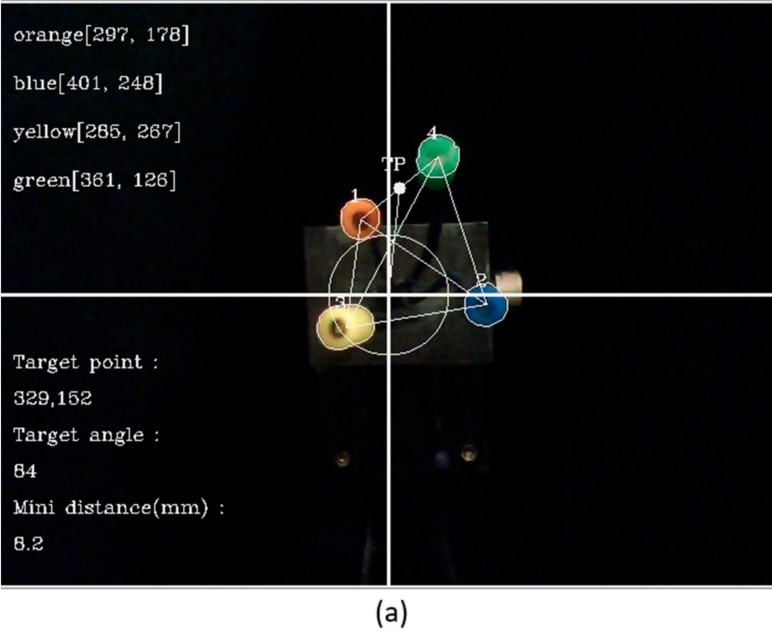

**Figure 21.** *Cont*.

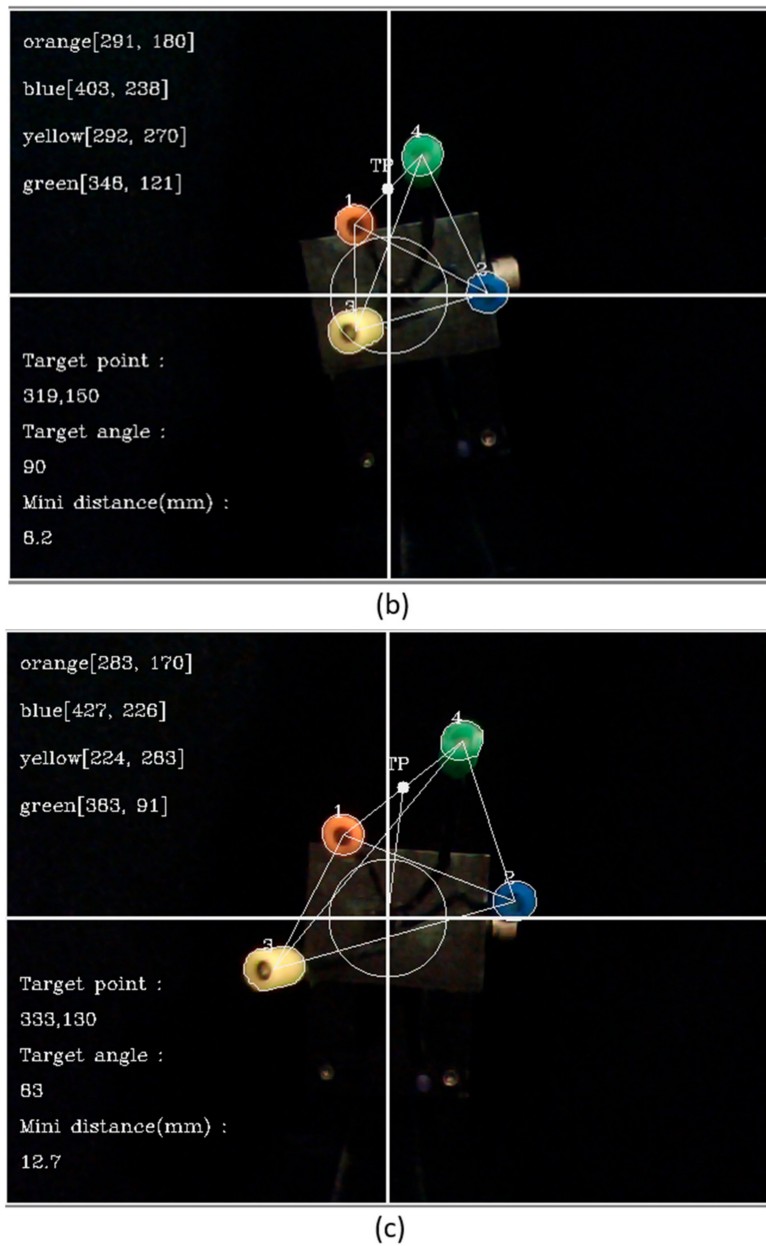

**Figure 21.** (**a**) Initial state: The distance between the four core wires is less than 8 mm. (**b**) Point TP is rotated to the center of the y-axis. (**c**) Success: The distance between the four core wires is more than 8 mm.

The experimental results are listed in Table 4. It can be observed from the table that the total success rate of the experiment is 94%; however, the probability of success of the first time is only 68%; thus, there is still scope for improvement in the experiment. For example, the friction between the separate jig and core wires determines the success rate of part of the wire separation operation. If the separate jig is changed to a smoother material or a higher-precision 3D printer is used to manufacture the separate jig, reducing the friction with the core wires will increase the success rate of the experiment. The time required for the first separation job is about 8 s and it outperformed the method proposed in [19] with a time reduction of 20%.

**Table 4.** Experimental results.

| Number of Experiments | The First Time Separate Job Succeeded (8 s) | The Second Time Separate Job Succeeded (16 s) | More than Three Times Separate Job Succeeded (24 s or More) | Separate Job Failed |
|---|---|---|---|---|
| 1–10 | 6 | 4 | 0 | 0 |
| 11–20 | 8 | 2 | 0 | 0 |
| 21–30 | 6 | 2 | 1 | 1 |
| 31–40 | 7 | 3 | 0 | 0 |
| 41–50 | 7 | 1 | 0 | 2 |
| Total | 34 | 12 | 1 | 3 |

The reasons for conducting repeated experiments can be summarized as follows:

1. If part of the wire is staggered, the wire can easily spring back because of its flexibility, and the wire can be separated after the second job.
2. When working, there is an opportunity to divide the other cores into the same quadrant; therefore, when the targets are separated, the distance between the other cores decreases slightly. In most cases, it can be performed with a second or more passes.

## 6. Conclusions

The currently used wire core separation module is expensive, and requires professional design of the mechanism. With the help of the new fixture design and visual feedback, the developed line separate module can quickly perform the wire separating operation. This economical and convenient separation module can overcome the shortcomings that exist in manually operated separation core wires and separation modules on the market, as well as increasing the achievability of fully automatic welding of core wires.

In addition, the traditional image processing method is easily affected by illumination changes in the parameter setting, which distinguishes each core wire by adjusting the parameters in the HSV color space. However, the colors of the core wires are similar; thus, they are difficult to distinguish. Therefore, this experiment added easily recognizable colors to the ends of the core wires. When the illuminance changes, the threshold parameters should be modified, which causes misjudgment in the actual operation. Therefore, to increase the accuracy of the identification of each core, the relative positions between the core wires can be segmented and obtained using deep-learning-based algorithms.

**Author Contributions:** Conceptualization, C.-C.H. and Y.-R.L.; methodology, C.-C.H.; software, Y.-R.L. and Y.-C.L.; validation, C.-C.H. and Y.-R.L.; writing—original draft preparation, Y.-R.L.; writing—review and editing, C.-C.H. All authors have read and agreed to the published version of the manuscript.

**Funding:** This research was supported by the National Science and Technology Council, Taiwan, under Grant no. 111-2218-E-002-031.

**Data Availability Statement:** Not applicable.

**Acknowledgments:** The authors thank the anonymous reviewers for their helpful suggestions.

**Conflicts of Interest:** The authors declare no conflict of interest.

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
