# Peer review of "Digital-Twin-Based Flexibility Wires of a Circular Connector Automatic Disassembly Process"

_robotics, doi:10.3390/robotics11060120_

Round 1

Reviewer 1 Report

This paper designs a separate novel module for cable core wires. A single-axis robot and a stepper motor are used to automatically separate the four core wires, which can potentially accelerate the subsequent welding work. It is interesting, but the writing can be improved before publication. Comments as follow:

  1. The first paragraph is too long, it's hard to read. The introduction section cites many references, but the necessity is not specified for this research.
  2. Page 6 "Owing to the flexibility…", The description of the experimental results in Table 2 and Figure 8 is a bit simple and needs to be more specific and rigorous. For example, "Experiments have shown that increasing the hole spacing can better separate the wire from its shape", Table 2 does not strictly prove this conclusion because E and F do not have a single variable.
  3. Figure 7 on page 5, the image of simulation results is not clear enough, and the parameters in the upper left corner of each picture cannot be seen clearly.
  4. Figure 16 on page 13 shows that the values of Mini distance(mm) in Figures a) and b) seem no different. Both are 8.2.
  5. Please pay attention to some formatting issues, including the cross-page display of the title of Figure 1, the branch display of equations 1) and 2) on page 9, the font of Table 3, and the layout of images and tables in the whole article.

Reviewer 2 Report

The literature seems to be too shallow, there exists numerous literature on the DT & Disassembly process of Electric/Electronic equipment in the context of Industry 4.0.

Negri, Elisa, Luca Fumagalli, and Marco Macchi. "A review of the roles of digital twin in CPS-based production systems." Procedia manufacturing 11 (2017): 939-948.

Bahubalendruni, M. V. A., and Vara Prasad Varupala. "Disassembly sequence planning for safe disposal of end-of-life waste electric and electronic equipment." National Academy Science Letters 44.3 (2021): 243-247.

In addition, the authors can collect some recent relevant literature using the keywords “disassembly prediction in industry 4.0; digital twin for disassembly process; augmented reality based product assembly; virtual sensors for digital twin; System modeling in DT”

The virtual twin/digital twin necessity for this process must be discussed.

The efficiency of the system/profit through the proposed process should be quantified.

A comparative assessment with reference to the existing methodology is required.

Visualizing the virtual model/process (product /disassembly process) in the augmented space is missing for a DT.

Provide units for the parameters in the tables. Use the axis system wherever applicable.

Reviewer 3 Report

This manuscript addresses an interesting technical subject. The scientific contribution is low since the design and its optimization is carried out only running simulations with a finite element program. It is valuable that experiments have been carried out.

In order to this manuscript merit publication:

1) The steps of Section 3.4 must be explained much better: equations (1), (2) and (3) are incomprehensible, what is the meaning of "the target point"? (it has not been defined previously to line 205), what is "separate feature" in line 207?. In general, all this section a much more detailed explanation.

 2) Expressions are introduced in an incorrect way: a reference to a equation should not appear prior to its definition, the labels of equations should be introduced as (x) in the formulas. Moreover, most of the variables that appear in the formulas are not defined.

3) I would appreciate some kind of matematical analysis that could help in the design of this tool, e.g., the deduction of a relationship between the friction, or the flexibility of the wires, and  the dimension of the tool cone.

Round 2

Reviewer 1 Report

The author has answered the questions.

Reviewer 2 Report

The manuscript is well revised, No further suggestions/recommendations.

All the best!.

Reviewer 3 Report

My comments have been adequately addressed. This manuscript can be published.